# Defect-Rich Monolayer MoS_2_ as a Universally Enhanced Substrate for Surface-Enhanced Raman Scattering

**DOI:** 10.3390/nano12060896

**Published:** 2022-03-08

**Authors:** Shiyu Sun, Jingying Zheng, Ruihao Sun, Dan Wang, Guanliang Sun, Xingshuang Zhang, Hongyu Gong, Yong Li, Meng Gao, Dongwei Li, Guanchen Xu, Xiu Liang

**Affiliations:** 1Key Laboratory for High Strength Lightweight Metallic Materials of Shandong Province (HM), Advanced Materials Institute, Qilu University of Technology (Shandong Academy of Sciences), Jinan 250014, China; sysun0313@163.com (S.S.); srh1645@163.com (R.S.); Wangdan1910@163.com (D.W.); 13793994223@163.com (G.S.); xszhang@qlu.edu.cn (X.Z.); hygong@sdas.org (H.G.); yongli@sdas.org (Y.L.); mgao@sdas.org (M.G.); 2College of Materials Science and Engineering, Fuzhou University, Fuzhou 350108, China; jyzheng@fzu.edu.cn

**Keywords:** MoS_2_, surface defects, surface-enhanced Raman scattering, photoinduced charge transfer, metallic

## Abstract

Monolayer 2H-MoS_2_ has been widely noticed as a typical transition metal dichalcogenides (TMDC) for surface-enhanced Raman scattering (SERS). However, monolayer MoS_2_ is limited to a narrow range of applications due to poor detection sensitivity caused by the combination of a lower density of states (DOS) near the Fermi energy level as well as a rich fluorescence background. Here, surfaced S and Mo atomic defects are fabricated on a monolayer MoS_2_ with a perfect lattice. Defects exhibit metallic properties. The presence of defects enhances the interaction between MoS_2_ and the detection molecule, and it increases the probability of photoinduced charge transfer (PICT), resulting in a significant improvement of Raman enhancement. Defect-containing monolayer MoS_2_ enables the fluorescence signal of many dyes to be effectively burst, making the SERS spectrum clearer and making the limits of detection (LODs) below 10^−8^ M. In conclusion, metallic defect-containing monolayer MoS_2_ becomes a promising and versatile substrate capable of detecting a wide range of dye molecules due to its abundant DOS and effective PICT resonance. In addition, the synergistic effect of surface defects and of the MoS_2_ main body presents a new perspective for plasma-free SERS based on the chemical mechanism (CM), which provides promising theoretical support for other TMDC studies.

## 1. Introduction

Two-dimensional transition metal dichalcogenides (TMDCs) have been very popular semiconductor materials with potential applications in many fields due to their unique structural and physicochemical properties [1,2,3,4,5]. Typically, semiconductor monolayers of MoS_2_ are of great interest for applications in electronic sensor devices, energy storage applications, catalytic fields, and composite materials. Additionally, MoS_2_ monolayers are substrates for potential applications of surface-enhanced Raman scattering (SERS) based on molecular detection, which strongly depends on the electronic and optoelectronic properties of monolayer MoS_2_ [6]. SERS is a nondestructive and useful quantitative analysis technique that is widely used because of its low cost, ease of synthesis, excellent optical properties, and biocompatibility [7,8,9,10,11,12]. The core of SERS research is the preparation of substrates [10,13,14,15,16], which requires a perfect balance between enhancement capability, homogeneity, stability, and economy. SERS substrate materials are enhanced by different mechanisms, which are the electromagnetic mechanism (EM) and the chemical mechanism (CM) [17,18,19]. The EM originates from localized surface plasmon resonance generated by metal nanostructures. The local amplified “hot spot” generated by surface plasmon resonance (SPR) is usually composed of noble metal nanoparticles (Au, Ag, or Cu), which have high SERS activity and detection sensitivity [20,21,22,23,24]. There is no substitute for precious metals, as SERS detection substrates, but they have the disadvantages of high cost and instability. In addition, there are problems of biocompatibility and chemical interactions between the detection molecule and the substrate in the process, and these problems seriously limit the market application of precious metal SERS substrates. The CM is mainly a charge transfer (CT) generated by the interaction between the chemistry of the detection molecule and the substrate. The CM is considered the main enhancement mechanism for 2D materials, such as graphene [25,26] and TMDCs [27,28]. The Raman signal generated by the nonmetallic SERS substrate represented by MoS_2_ is a combination of photoinduced charge transfer (PICT) between the dye molecule and the substrate as well as local dipoles resulting from changes in molecular symmetry [29,30,31]. Such substrates not only have higher SERS homogeneity but also better chemical stability and biocompatibility [32]. In addition to the unique enhancement mechanism, the surface of the 2D material allows for uniform chemisorption of dye molecules, which is essential for reproducible signal collection and practical applications [19,33]. However, the detection performance of monolayer MoS_2_ as a SERS substrate is lower than the enhancement brought by noble metals due to its high charge recombination rate, low electrical conductivity, limited catalytic active sites, and weak charge transfer. The precise control and modulation of the electronic and optoelectronic properties of MoS_2_ can improve its versatility. According to Fermi’s golden rule, the electron leap probability of the SERS process is linearly related to the density of states near the Fermi energy level [9,27,34]. In recent years, many tuning strategies have been applied to increase the density of states (DOS) in the semiconductor bandgap to in turn improve the PICT efficiency [27]. A range of techniques for fabricating defects has been developed, including phase transitions [35], plasma treatment [36], and oxygen doping [32], among others [37]. Notably, these defect-enhanced methods for improving the SERS performance of 2D materials are complex and extremely limited, if not negative. For this reason, it is necessary to develop a simple and widely applicable defect engineering strategy based on monolayer MoS_2_ that does not change the overall material properties while achieving high electron density [2,38]. In addition, Warner et al. provided a new idea for surface defects by demonstrating that increasing the width of linear defects leads to reconstructed nanoscale regions of monolayer MoS_2_, which density functional theory (DFT) shows to exhibit metallicity [39].

Here, for the first time, we use experiments and theory to investigate the Raman enhancement effect of monolayer MoS_2_ before and after the presence of surface defects. In this paper, large-scale, highly crystalline samples were prepared under the precise control of the chemical vapor deposition (CVD) method with sulfur powder and electroplated molybdenum foil as precursors of the monolayer MoS_2_ and with SiO_2_/Si as substrates [40,41,42]. Indeed, due to the strong fluorescence background and inefficient CT, it is difficult to visualize the SERS signal in perfect MoS_2_ monolayers, especially when the surface contains low concentrations of detected molecules. In contrast, monolayers of MoS_2_ containing a small number of defects are produced by short-term, high-temperature etching in a constant low-pressure environment with only a small amount of air. Monolayer MoS_2_ containing etched regions exhibits the unique advantage of abundant DOS near the Fermi energy level and more intense CT and PICT efficiency. This strategy is broad, with a wide range of detection molecules being detected and reaching limits of detection (LODs) of less than 10^−8^ M. Theoretical calculations further confirm that monolayer MoS_2_ in the etched region possesses a narrower bandgap and exhibits metallic properties. Metallic MoS_2_ has abundant DOS near the Fermi energy level and strong interactions with detection molecules, making it ideal as a promising enhanced substrate for SERS applications.

## 2. Materials and Methods

### 2.1. Synthesis of Monolayer MoS_2_

The monolayer MoS_2_ nanosheets are prepared through a CVD system using sulfur and molybdenum sources as precursors. Prepare the electrolyte with Na_2_SO_4_ (Aladdin Reagents, 99.0%), NaF (Aladdin Reagents, 99.93%), and H_2_C_2_O_4_ (Macklin, 99.0%) according to a specific ratio, and perform anodizing at room temperature with a voltage of 0.4~0.6 V and a time of 30 min. In the CVD system, the sulfur powder (Macklin, 99.99%) in the corundum boat is placed upstream of the pipeline, and the electroplated molybdenum foil on the SiO_2_/Si wafer is placed in the center of the furnace. A separate heating zone is the heat source for sulfur sublimation. When the heating zone reaches 150 °C, the temperature of the molybdenum source just reaches 850 °C, and all react in the Ar atmosphere for 15 min. When the temperature in the furnace drops to 50 degrees Celsius, the heating of the S source is stopped. Keep the Ar atmosphere until the system reaches room temperature.

### 2.2. Synthesis of Etched MoS_2_

The perfect monolayer MoS_2_ is selected to prepare defects to reduce experimental errors. MoS_2_ is placed in the center of the furnace with one end sealed, and the other end is a continuously working vacuum oil pump. An etching environment that is almost isolated from the outside world with only a small amount of air is created at a low vacuum of 0.02 Torr. In order to maintain the crystal structure of MoS_2_, the etching temperature is controlled below the growth temperature, and the reaction time is within a few minutes.

### 2.3. Materials Characterization

The morphology of molybdenum foil was characterized by SEM (S4800, Hitachi, Tokyo, Japan). A three-electrode system and a CHI 760D (Shanghai, China) electrochemical workstation were used for the electrochemical experiment. Optical images were captured with an Olympus BX 53 M microscope (Tokyo, Japan). AFM from a Bruker bioscope resolve system and silicon cantilevers from nano sensors were used for intelligent mode operation. Raman spectra were recorded from a Horiba JY iHR550 (Tokyo, Japan) system with an excitation wavelength at 532 nm. During measurements, the laser beam was focused on a spot with a 1 μm diameter by a microscope objective with a magnification of 50×, and the acquisition time was set to 5 s.

### 2.4. SERS Measurements

Rhodamine 6G (R6G), methylene blue (MB), and crystal violet (CV) dyes were used as Raman probe molecules to first verify the SERS properties of etched MoS_2_ substrate. Briefly, 15 μL of R6G, CV, and MB probes with different concentrations (10^−4^–10^−9^ M) were consecutively dropped on the surface of the etched substrate. The laser power on the sample was 50 mW with a 5 s exposure time under 532 nm laser excitation. We calculated the enhancement factor (EF) values using the following equation:EF=ISERSNSERSINRNNR
where I_SERS_ and I_NR_ refer to the Raman intensity of probe molecules in the SERS and Raman spectra, and N_SERS_ and N_NR_ are the estimated molecule number under laser excitation for SERS and the molecule number for the reference sample (solid), respectively.

### 2.5. DFT Calculation Details

In the framework of the density functional theory (DFT), the generalized gradient approximation and projector broadening plane wave method proposed by Perdew, Burke, and Ernzerhof was used in the framework of density generalized function theory at a kinetic energy cutoff of 400 eV, as implemented in the Vienna ab initio simulation package (VASP 5.4.4) [43,44]. The Brillouin zone of the surface unit cell was sampled by Monkhorst–Pack (MP) grids, with a different k-point mesh for MoS_2_ structure optimizations. The MoS_2_ surface was determined by a 2 × 2 × 1 Monkhorst−Pack grid. The convergence criterion for the electronic self-consistent iteration and force was set to 10^−5^ eV and 0.01 eV/Å, respectively. A 6 × 6 supercell of the MoS_2_ surface, including one layer, was constructed to model the MoS_2_ catalyst in this work. A vacuum layer of 15 Å was introduced to avoid interactions between periodic images.

## 3. Results and Discussion

Controllable preparation of the material is the key to achieving practical applications of single-molecule thick MoS_2_. Figure 1a examines a series of advances centered on monolayer MoS_2_, including preparation, evaluation of MoS_2_ containing defects, and characterization of various SERS properties. Sublimable sulfur powder and molybdenum foil were used to prepare monolayer MoS_2_ in a CVD system, which was an improvement demonstrated in previous works [40,41]. The metallic molybdenum foil was electroplated and treated on an electrochemical workstation. Its activity was enhanced to be more favorable for the reaction (Appendix A). Specifically, the growth process is such that the CVD is designed with two different temperature-controlled zones, and when Mo reaches the MoS_2_ growth temperature, the S vapor rapidly fills the entire system (Appendix A). Both vapors collect simultaneously on the SiO_2_/Si substrate and rapidly nucleate growth with a size of approximately 30 μm, as shown in Figure 1b,e. 

Here, based on the above monolayer MoS_2_, a simple and practical defect preparation strategy is proposed: a thermal etching process in a low-vacuum system. Given the complex variation of external conditions, a vacuum pressure oil pump maintains the internal pressure in a closed environment at a low vacuum of 0.02 Torr and externally applies a high temperature of ~800 °C below the growth temperature (Appendix A), which creatively generates abundant defects in a short time. The etching conditions were chosen for the case of moderate etching as a result of several experiments (Appendix A). The extended pumping time before etching and the presence of excess S monomer on the SiO_2_/Si substrates with a monolayer of MoS_2_ serve to minimize the effect of oxidizing gases in the confined system. This approach is very clever, neither destroying the structure of the monolayer MoS_2_ nor producing only the absence of Mo and S atoms. These subtle changes are indistinguishable under optical microscopy (Figure 1b,c), but can be observed under atomic force microscopy (AFM), as seen in Figure 1e,f. The etched MoS_2_ surface becomes rough and the corresponding impurities on the SiO_2_/Si substrate are largely removed (Appendix A). A more objective SERS detection property is that the MoS_2_ surface containing defects adsorbs more dye molecules, which is reflected in the optical photographs and AFM in Figure 1d,g. Strong chemisorption enhances the connection between the substrate and the dye molecules, which enhances the CT and coupling effects. Relatively, the monolayer MoS_2_ surface adhesion is small and uneven (Appendix A).

MoS_2_ underwent a large change during the etching process. The Raman spectrum was obtained using a 532 nm laser line with a spot size of approximately 1 μm. From the Raman spectra in Figure 2a, two characteristic MoS_2_ peaks are located at ~383.0 cm^−1^ and ~401.9 cm^−1^, which are associated with in-plane (E^1^_2g_) and out-of-plane (A_1g_) vibrational modes, respectively [45]. The peak spacing between the two characteristic peaks is ~19 cm^−1^, which confirms the monolayer structure of MoS_2_. After the appearance of surface defects in the MoS_2_, the two Raman characteristic peaks are shifted, and the peak spacing becomes wider. It is worth mentioning that the characteristic peak A_1g_ in the etched MoS_2_ Raman spectrum is shifted approximately 1 cm^−1^ in the high wavenumber direction due to a small amount of air remaining in the confined environment reacting with the MoS_2_. 

The photoluminescence spectra (PL) spectra show that defects can affect the bandgap of the semiconductor and cause fluorescence bursts in MoS_2_ (Appendix A). The calculated energy band structure yields a bandgap of approximately 1.70 eV for monolayer MoS_2_ (Appendix A), whereas MoS_2_ at the defect exhibits strong metallic properties of only 0.07 eV (Figure 2b). X-ray photoelectron spectroscopy (XPS) was used to demonstrate the chemical state of the monolayer MoS_2_ after etching, by comparing commercial MoS_2_ powder with defect-less monolayer MoS_2_. The in situ semi-quantitative analysis by XPS shows that the content of Mo in defect-rich MoS_2_ decreases from 2.48% to 1.74% and that S decreases from 3.78% to 2.01% relative to the unetched sample. Both the Mo 3d and S 2p spectra of the MoS_2_ can be fitted well, indicating the elemental composition and valence states in Figure 2c,d. The binding energy peaks at 229.6 eV and 232.7 eV for monolayer MoS_2_ are caused by Mo^4+^ 3d_5/2_ and 3d_3/2_. The peak at 233.8 eV corresponds to Mo^6+^ 3d_5/2_, and the peak at 235.9 eV corresponds to Mo^6+^ 3d_3/2_. The Mo^6+^ is due to a small number of oxidation peaks due to the great difference between the CVD system and the external environment during the preparation process and due to the inverse concentration gradient flow of O_2_ in the air. The monolayer MoS_2_ is etched at a high temperature and a low air concentration, the Mo 3d XPS spectral peak shifts in the direction of the low binding energy, the Mo^4+^ 3d_3/2_ content decreases, and the Mo^6+^ content disappears completely but is replaced by Mo^5+^ 3d_5/2_ at 232.7 eV and Mo^5+^ 3d_3/2_ at 235.6 eV. The presence of S 2p_1/2_ and S 2p_3/2_ spin-orbit bimodal peaks in the monolayer MoS_2_ is related to Mo-S bonding. Curiously, S^6+^ appears in the MoS_2_ monolayer due to the residual SO_4_^2−^ ion during the plating of the molybdenum foil used for the preparation of the monolayer MoS_2_ precursor by the CVD method. The complete disappearance of S^6+^ after the etching process is one of the reasons why the S 2p XPS spectrum also moves toward low binding energy. Comparing the Mo 3d and S 2p of commercial MoS_2_ powders, no significant Mo-O and S-O bonds appear after the etching reaction. The enhancement of SERS performance due to surface defects is mainly due to Mo and S defects rather than O_2_. The binding energy of the XPS spectra of O1s etched with MoS_2_ did not change compared to that of monolayer MoS_2_ (Appendix A). For commercial MoS_2_, no Mo-O bond exists before etching, so there is no effect of oxygen defects even after the reaction.

To investigate the SERS effect of MoS_2_ on SiO_2_/Si substrates, rhodamine 6G (R6G) was first used as a probe molecule. A series of R6G solutions were prepared from 10^−8^ M to 10^−4^ M to investigate the SERS sensitivity of 2D etched monolayer MoS_2_. There was a significant difference in the SERS behavior between monolayer MoS_2_ without defects and the etched MoS_2_ (Figure 3a). The 10^−4^ M concentration of R6G solution added dropwise to the clean SiO_2_/Si substrate did not result in any detectible SERS signal. At the same concentration, the Raman signal of the unetched monolayer MoS_2_ grown on SiO_2_/Si as a substrate for SERS detection was barely detectable due to the interference of the fluorescence background. Monolayer MoS_2_ containing defects exhibited exaggerated SERS performance. The characteristic R6G peaks located at 611, 773, 1360, and 1640 cm^−1^, could be easily observed on the etched MoS_2_, which were assigned to the C−H ring in-plane bending of the xanthenes skeleton, C−H out-of-plane bending, CH_3_ bending, and C–C stretching vibration modes, respectively [9,30]. The chemisorption caused a certain degree of distortion and polarization of the molecular structure, which shifted the vibrational band compared to the normal Raman spectrum of the R6G powder. When the concentration of R6G reached 10^−8^ M, it was still detectable that the monolayer MoS_2_ substrates contain defects. The data show an enhancement factor (EF) of 10^5^, which is comparable to that of noble metal-based SERS substrates (Appendix A). 

To further explore the wide applications of defect-containing MoS_2_ for SERS, tests were performed with different concentrations of CV and MB molecules under the same test conditions. The Raman peaks of methylene blue (MB) molecules located at 1625 and 1398 cm^−1^, and crystal violet (CV) molecules located at wavenumbers of 918, 1177, 1376, 1587, and 1620 cm^−1^ were detected (Appendix A) [17]. The LODs of both molecules can reach 10^−8^ M, which is superior to most semiconductor SERS substrates. The monolayer MoS_2_ grown on the same SiO_2_/Si exhibited a high onset of SERS signal measured after the etching reaction. When CV was the detection molecule (Appendix A), the relative standard deviations (RSD) of Raman peaks at 1620 cm^−1^ were calculated to be 6.2%. This result is due to the homogeneity of the defect generation process by etching and, to a lesser extent, the relatively strong and homogeneous interaction between the dye molecule and the etched MoS_2_ surface. The stability of SERS measurements is also a key issue for application. Etched MoS_2_ samples with deposited R6G (10^−5^ M) were exposed to a typical laboratory temperature atmosphere for 9 months for in situ Raman measurements (Figure 3c). The results show that the Raman intensity of the characteristic R6G (10^−5^ M) peak at 1640 cm^−1^ on the substrate still maintains 20% SERS activity after 9 months (Appendix A). This period is rare among the reported SERS substrates and confirms the excellent stability of the etched substrate under air exposure [9]. The excellent detection performance of etched monolayer MoS_2_ contributes to the establishment of a general method for the application of a wide range of semiconductor SERS sensors in realistic analytical scenarios.

To further clarify the enhancement mechanism in-depth for the excellent SERS activity of 2D etched monolayer MoS_2_, systematic investigations have been conducted by theory simulations. DOS simulations further show that the Fermi energy level of the etched monolayer MoS_2_ is elevated due to the presence of surface defects. Relatively, the free electron and charge densities of etched MoS_2_ near the Fermi energy level are increased. These changes are expected to have important implications for SERS performance due to increased PICT jumps as well as substrate-molecule interactions. The highest occupied molecular orbital (HOMO) and lowest-unoccupied molecular orbital (LUMO) levels of R6G were −3.46 eV and −6.28 eV, respectively (Figure 4b). 

Based on the above experimental results and theoretical analysis, possible charge transfer mechanisms of multiple dye molecules were further considered to better understand the differences between monolayer MoS_2_ and defect-containing monolayer MoS_2_. Monolayer MoS_2_ absorbs more dye molecules after etching than before etching, but this is not enough to greatly enhance the SERS performance. The electron transfer of monolayer MoS_2_ under laser irradiation at 532 nm (2.33 eV) can be described by a two-step process. Process I is a molecular leap (μ_mol_), in which electrons are excited at the HOMO energy level to the LUMO energy level of R6G, leaving holes in the HOMO. Process II, the PICT leap process (μ_PICT_) based on the CM mechanism, occurs between the substrate and the R6G molecule at the energy of the applied laser, which involves two processes: from the valence band (VB) of the monolayer MoS_2_ to the HOMO of R6G, and from the HOMO of R6G to the conduction band (CB) of the monolayer MoS_2_. The process energies of these two PICT jumps are 0.65 eV and 1.01 eV, which are far below the laser light energy of 2.33 eV. In the R6G/etched MoS_2_ system, in addition to the two PICT processes of VB-LUMO and HOMO-CB between R6G and the monolayer MoS_2_, there is also a coupling of etched MoS_2_ to R6G. The CB and VB of etched MoS_2_ are located at −3.79 eV and −3.86 eV, respectively (Appendix A). The PICT jumps from the R6G HOMO to the etched MoS_2_ Fermi energy level and then from the etched MoS_2_ Fermi energy level to the LUMO of R6G, which in turn generates a wide range of charge transfer resonance energies. The resonance gains intensity from the molecular leap through vibrational coupling, which greatly amplifies the polarization tensor of the probe. In general, the polarization tensor is expressed as α = A + B + C, where A is associated with the molecular resonance and B and C represent the two charge-transfer resonances [46]. In addition to the charge transfer resonance, molecular resonance at 532 nm excitation also increases the cross-section of the Raman scattering probe, whereas no detectable SERS signal is collected at the 633 nm laser wavelength (Appendix A) [47]. Second, molecular fluorescence background burst is also considered as another manifestation of charge transfer in the R6G/etched MoS_2_ system. Based on the above points, defect-containing MoS_2_ exhibits stronger SERS activity than monolayer MoS_2_.

The CV and MB molecules on the MoS_2_ substrate with defects also greatly enhance the SERS performance due to the CM-based PICT leap (Appendix A) due to the stability (a) and its ratio (b) of 10^−5^ M R6G coated on the etched MoS_2_. The electron transfer process between CV and MB molecules and MoS_2_ is much more complicated. In addition to their μ_ex_, two PICT processes, VB-LUMO and HOMO-CB, occur between the two molecules and the unetched MoS_2_, as shown in Appendix A. In CV/etched MoS_2_ or MB/etched MoS_2_, the charge of the LUMO of CV or MB must jump further to the Fermi energy level of the etched MoS_2_. This process is advantageous, and metallic defect-containing MoS_2_ with abundant DOS near the Fermi energy level as well as strong interactions with analytes also enhance SERS properties. The known μ_mol_ energies are 1.79 eV and 1.80 eV for CV and MB molecules, respectively. These values are less than 2.17 eV, which is the μ_mol_ energy for R6G molecules. At the same excitation energy, the SERS performance of CV and MB molecules is nevertheless consistent with that of R6G, which may be caused by the weakening of the resonance energy between the substrate and the molecule.

The coupling and PICT modes between the three dye molecules and the etched MoS_2_ were investigated using density flooding theory. Side views of the electron density difference equivalence surface were used to interpret the CT direction more intuitively (Figure 4c and Appendix A). The blue area is used to indicate an increase in the charge density, and the red area is used to indicate a decrease in charge density. There is a general loss of electrons on the dye molecule and an overall electron-rich state in the MoS_2_. The electrons and holes generated by charge transfer are closely located around the defective MoS_2_ surface and the dye molecules, forming dipoles at the interface, and the dipole resonance excites the SERS effect of the dyes on the etched MoS_2_. Additionally, the presence of regions of increased and decreased electron density on both R6G, MB, and CV and monolayer MoS_2_ containing defects indicates that the charge transfer in the energy band structure is reasonable and that this defect preparation strategy is effective for the enhancement of the SERS performance of monolayer MoS_2_.

## 4. Conclusions

In summary, a strategy to prepare defects by etching was applied to monolayer MoS_2_. Defects are considered to be the absence of Mo and S atoms, which leads to changes in the MoS_2_ surface and energy band structure. The defects tend to be in a metallic state. This change results in a dramatic improvement in the SERS properties of monolayer MoS_2_. Substrates prepared from defect-containing monolayer MoS_2_ can not only detect a wide range of dye molecules but also maintain ultrahigh reproducibility and stability. The results of theoretical calculations show that the static coupling between the dye molecules and the monolayer MoS_2_ containing defects, the increase in the DOS near the Fermi level, and the enhancement of the charge transfer resonance make SERS detection more sensitive. Etched MoS_2_ is a new strategy for the molecular engineering of 2D materials and provides new ideas for SERS applications of other TMDCS materials, making it promising for both materials science and chemistry.

## Figures and Tables

**Figure 1 nanomaterials-12-00896-f001:**
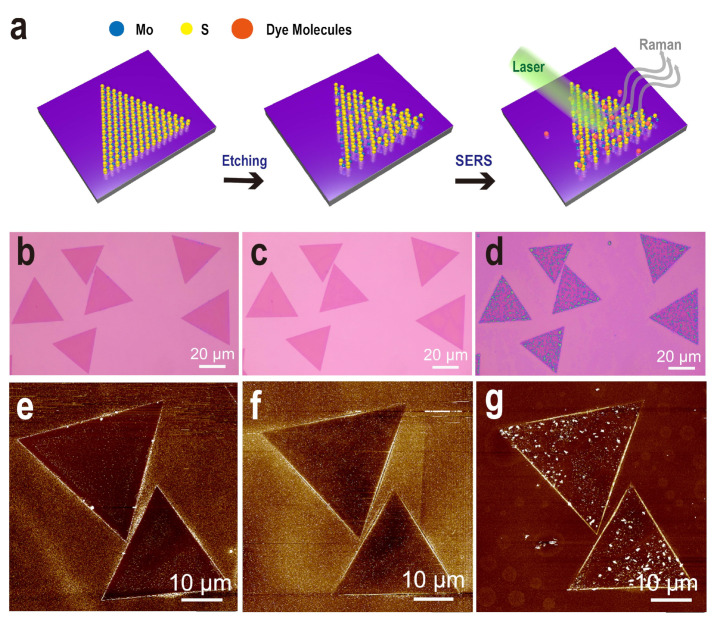
(**a**) Simple schematic illustrating the process of SERS detection after etching monolayer MoS_2_ grown on the same SiO_2_/Si substrate. Optical microscopy image (**b**–**d**) and AFM image (**e**–**g**) of the monolayer MoS_2_, the etched monolayer MoS_2,_ and the etched monolayer MoS_2_ with dye molecules added dropwise.

**Figure 2 nanomaterials-12-00896-f002:**
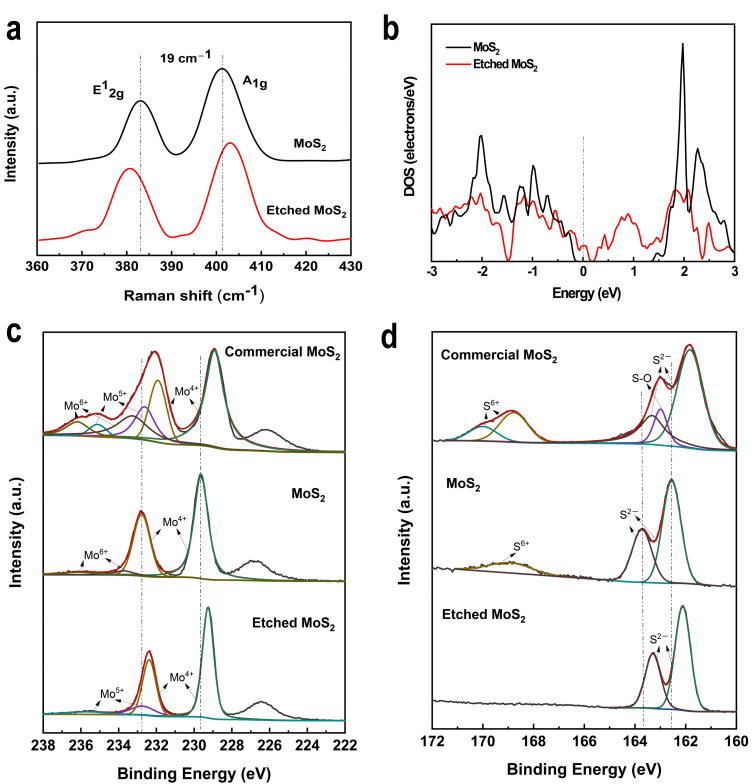
(**a**) Raman spectra and density of states of CVD grown MoS_2_ and etched monolayer MoS_2_. (**b**) Calculated band structures of etched MoS_2_ using the Fermi level as a reference. XPS data of commercial MoS_2_ powder, CVD grown monolayer MoS_2_, and etched monolayer MoS_2_ at the binding energies of (**c**) Mo 3d and (**d**) S 2p.

**Figure 3 nanomaterials-12-00896-f003:**
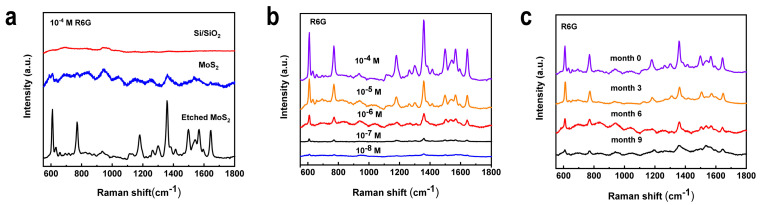
SERS measurements of rhodamine 6G (R6G) molecular probes on the etched monolayer MoS_2_ substrates: (**a**) Raman peaks of 10^−4^ M R6G; (**b**) Concentration-dependent SERS spectra from 10^−8^ M to 10^−4^ M; (**c**) The stability ratio of 10^−5^ M R6G coated on the etched MoS_2_.

**Figure 4 nanomaterials-12-00896-f004:**
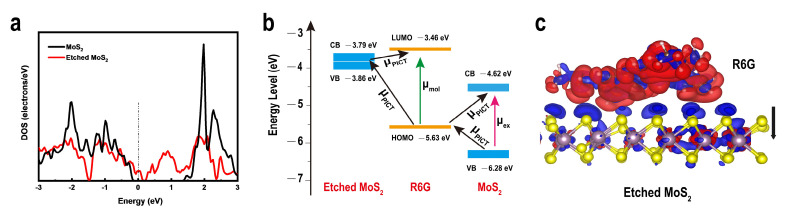
(**a**) Density of states of CVD grown MoS_2_ and etched monolayer MoS_2_. (**b**) Energy level diagram and charge transfer transitions in the diagram comparing the charge-transfer pathways in R6G/MoS_2_ and R6G/etched MoS_2_-MoS_2_. (**c**) Side views of the electron density difference isosurface for the R6G molecule absorbed on etched monolayer MoS_2_.

## Data Availability

Data are available on the request of the corresponding author.

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
