# Peer review of "Defect-Rich Monolayer MoS_2_ as a Universally Enhanced Substrate for Surface-Enhanced Raman Scattering"

_nanomaterials, 2022, doi:10.3390/nano12060896_

Round 1
Reviewer 1 Report
The work presents routine studies of Raman signal amplification on an SERS-active substrate. Monolayers of MoS2 with defects are chosen as such substrates, which is a scientific novelty. The presented experimental results are in good agreement with theoretical calculations, and the conclusions are not contradictory. Nevertheless, there is a question for the authors regarding the scheme proposed in Figure 4b. It is known that the absorption band of Rhodamine 6G overlaps the region around 532 nm and does not include the region around 633 nm. Why is 532 nm photon energy insufficient for direct excitation of Rhodamine 6G molecules?
Reviewer 2 Report
The present work reports on the surface enhanced Raman scattering (SERS) properties of defect MoS2 monolayers grown on Si/SiO2 substrate. The experimental Raman spectra, showing clear enhancement on defect layers are interpreted in the light of density functional theory (DFT) calculations for MoS2, which infer for a chemical mechanism (CM) of enhancement trough charge transfer (CT) to the enhanced electron DOS near the Fermy energy of the defect material.
The reported experimental enhancement factors (EF) and detection limits of 10-8M are quite promising and qualify the as prepared MoS2 defect monolayers as an useful SERS substrate material. The idea of using defect MoS2 and the reported results are original and may have a good impact among the practitioners in the field of SERS. However, in order to meet the high criteria of Nanomaterials, I suggest some revisions to the manuscript, before proceeding to eventual publication.
- First and foremost, authors should state clearly the nature of defects, created in the MoS2 layer through etching. Does “surfaced S and Mo atomics defects”, as mentioned in the abstract, mean atomic vacancies, add-atoms, or atypical atomic coordination? Authors certainly should present some reliable estimates of the concentration of defects.
- In the context of the first question, it is highly desirable to compare the SERS EF for MoS2 substrates with different defect concentration. Any correlation between EF and the defect concentration at a fixed R6G concentration could serve as solid evidence in favor of the mechanism proposed by the authors.
- How are estimated NSERS and NNR, as defined by the equation for EF on line 136?
- The DFT calculation details (section 2.5) are too scarce. First of all authors should describe in details the methodology of electronic structure calculations for the defect MoS2 layer. Usually defects are considered in a supercell of the parent material in order to make use of the periodic boundary conditions. Is it the case in the present study? If so, what is the size of the supercell, and once again, what is the concentration of the defects?
- The 2 x 2 (x1) k-point mesh in DFT calculations seems two small in order to produce reliable results concerning electronic structure since it includes only the center and the edge of the BZ. Does” 2 x 2” refer to a supercell (if considered) or to the BZ of the pure MoS2 material?
- From the very beginning authors exclude the electrodynamic mechanism (EM) of SERS enhancement the system under study. On the other hand, one may speculate that the localized defects of metallic-like behavior may support localized plasmonic resonances. Could authors comment in more details the relevance or irrelevance of EM in the present case.
- Authors use several abbreviations without defining them upon the first use, like DFT, EF (enhancement factor), R6G (rhodamine 6-G), VC, MB. Although, these abbreviations may seem evident , the good scientific style assumes proper definition, which could help the readers who are not specialists in the field.
Reviewer 3 Report
Paper: [nanomaterials-1615311] Defect-rich monolayer MoS2 as a universally enhanced sub-2 strate for surface-enhanced Raman scattering
The authors described experimental and theoretical data of SERS signal improvements after surfaced S and Mo atomics defects are fabricated on a monolayer MoS2 with a perfect lattice. A enhancement factor of 105 is reported with a detection limit of 10-8 M R6G. Overall the manuscript is well written and comprehensive data. It is recommended for publication with the minor comment of using paragraphs to break up the text under each sub-section of the “Results & Discussion” section. This will provide more clarity for the readers.
Round 2
Reviewer 2 Report
Authors have addressed the questions and suggestions in my previous report in exhaustive way.. Corresponding modifications have been made in the revised manuscript, like a more detailed description of the DFT calculation methodology. Thus, I am persuaded that the manuscript can be published in its present form.